# Seroprevalence of SARS-CoV-2–Specific Antibodies in Cancer Patients Undergoing Active Systemic Treatment: A Single-Center Experience from the Marche Region, Italy

**DOI:** 10.3390/jcm10071503

**Published:** 2021-04-04

**Authors:** Luca Cantini, Lucia Bastianelli, Alessio Lupi, Giada Pinterpe, Federica Pecci, Giovanni Belletti, Rosa Stoico, Francesca Vitarelli, Marco Moretti, Nicoletta Onori, Riccardo Giampieri, Marco Bruno Luigi Rocchi, Rossana Berardi

**Affiliations:** 1Rossana Berardi, Clinical Oncology, Università Politecnica delle Marche, A.O.U. Ospedali Riuniti, Via Conca 71, 60126 Ancona, Italy; lucacantini.med@gmail.com (L.C.); lucia.bastianelli.ps@gmail.com (L.B.); lupialessio2@gmail.com (A.L.); giadapinterpe@gmail.com (G.P.); peccifede91@gmail.com (F.P.); giobellet94@gmail.com (G.B.); rosa.stoico@ospedaliriuniti.marche.it (R.S.); francesca.vitarelli@ospedaliriuniti.marche.it (F.V.); riccardo.giampieri81@gmail.com (R.G.); 2SOD Medicina di Laboratorio, A.O.U. Ospedali Riuniti, 60126 Ancona, Italy; Marco.Moretti@ospedaliriuniti.marche.it (M.M.); nicoletta.onori@ospedaliriuniti.marche.it (N.O.); 3Department of Biomolecular Sciences, University of Urbino Carlo Bo, 61029 Urbino, Italy; marco.rocchi@uniurb.it

**Keywords:** COVID-19, SARS-CoV-2–specific antibodies, cancer patients, seroprevalence

## Abstract

Severe acute respiratory syndrome coronavirus 2 (SARS-CoV-2) seroprevalence in cancer patients may vary widely dependent on the geographic area and this has significant implications for oncological care. The aim of this observational, prospective study was to assess the seroprevalence of SARS-CoV-2 IgM/IgG antibodies in solid cancer patients referred to the academic institution of the Marche Region, Italy, between 1 July and 26 October 2020 and to determine the accuracy of the rapid serological test. After performing 3767 GCCOV-402a rapid serological tests on a total of 949 patients, seroconversion was initially observed in 13 patients (1.4%). Ten (77% of the total positive) were IgG-positive, 1 (8%) were IgM-positive and 2 (15%) IgM-positive/IgG-positive. However, only 7 out of 13 were confirmed as positive at the reference serological test (true positives), thus seroprevalence after cross-checking was 0.7%. No false negatives were reported. The kappa value of the consistency analysis was 0.71. Due to rapid serological test high false positive rate, its role in assessing seroconversion rate is limited, and the standard serological tests should remain the gold standard. However, as rapid test negative predictive value is high, GCCOV-402a may instead be useful to monitor patient immunity over time, thus helping to assist ongoing vaccination programs.

## 1. Introduction

On 11 March 2020 Coronavirus disease 19 (COVID-19) outbreak was declared pandemic by the World Health Organization (WHO). First published data suggested that patients with active malignancy might be at increased risk of both being infected by severe acute respiratory syndrome coronavirus 2 (SARS-CoV-2) and developing COVID-19 complications because of their underlying disease and their immunosuppressive status due to oncologic treatments [1]. However, the impact of COVID-19 on cancer patients is still not clearly defined. A Chinese evaluation estimated that, among patients admitted to Radiotherapic and Oncological Departments of Zhonghan Hospital of Wuhan University (China), 0.79% of cancer patients developed symptoms related to SARS-CoV-2 infection. This percentage was higher compared to disease prevalence in the general population (OR 2.31, 95% IC 1.89–3.02) [2]. Conversely, an analysis conducted by the Antwerp University group suggested that SARS-CoV-2 seroprevalence was lower in cancer patients compared to healthcare professionals (8.5% vs. 16% respectively) [3]. A large Chinese study among people with COVID-19 disease demonstrated that oncological patients had higher risk of severe outcomes (including severe pneumonia or death) due to SARS-CoV-2 infection than non-cancer patients [4]. However, mortality among cancer patients affected by SARS-CoV-2 infection varies greatly according to different studies [5,6,7]. Therefore, current evidence has not clearly estimated the risk of SARS-CoV-2 infection and severity in cancer patients, nor even specific predictive/prognostic factors have been found yet [8]. 

Nasopharyngeal swab (NPS) is the gold standard diagnostic test to confirm SARS-CoV-2 infection, followed by SARS-CoV-2 RNA detection by reverse-transcription PCR (RT-PCR). Although not representing a reliable screening method as adaptive immune response takes some time to develop antibodies, serological tests detecting SARS-CoV-2 IgM and IgG antibodies can represent a useful tool to help determine COVID-19 seroprevalence thanks to easy execution, fast reporting, good diagnostic sensitivity and specificity [9,10]. The detection of IgM/IgG SARS-CoV-2 antibodies allows for an estimation of the population developing antibodies that could potentially be protected against Coronavirus disease. A survey conducted by the Italian Ministry of Health and the Italian National Institute of Statistics (ISTAT) with the collaboration of the Italian Red Cross (CRI) showed that, at data cut-off (15 July), 2.5% of the Italian population had developed antibodies to SARS-CoV-2 and that 27.3% of people who developed antibodies never had symptoms. According to this study, seroprevalence among the population of the Marche Region was estimated at 2.7% [11]. Little is known about the detection rate of SARS-CoV-2 antibodies in oncologic patients. Seroprevalence in this frail population differs mostly according to the geographic areas [12,13]. The aim of this observational study was to assess the seroprevalence of IgM/IgG antibodies in cancer patients undergoing systemic treatment at Clinica Oncologica Azienda Ospedaliero-Universitaria Ospedali Riuniti di Ancona—Università Politecnica delle Marche (UNIVPM), as per assessed by rapid and serological tests. By this, the study also aimed at assessing the total agreement between the rapid serological test and the standard serological examination. 

## 2. Materials and Methods

### 2.1. Study Design and Study Population

This study was a monocentric, observational, prospective study to investigate the seroprevalence of COVID-19 antibodies in patients undergoing medical anticancer treatment in the Clinica Oncologica UNIVPM between July and October 2020. All consecutive patients who received systemic anticancer treatment (administered intravenously or subcutaneously/intramuscularly) at the UNIVPM outpatient and inpatient clinic between 1 July and 26 October 2020 were asked to be included in the TACCO (“serological Test detecting Anti-sars-Cov-2 at Clinica Oncologica”) procedure. Patients receiving oral anticancer treatments were included only if oral therapy was given as part of a treatment regimen that also included intravenous or subcutaneous therapy. Patients receiving systemic treatment for hematological cancers were excluded from the procedure. Patient and tumor characteristics, as well as data on cancer treatment and previous COVID-19 history, were collected by patients’ interview and medical records review. To evaluate the seroprevalence of IgG and IgM against SARS-CoV-2, all enrolled subjects were first tested with the COVID-19 IgG/IgM Rapid Test Cassette GCCOV-402a (Zhejiang Orient Gene Biotech Co., Ltd., Zhejiang, China), which is a solid phase immunochromatographic assay for the rapid, qualitative and differential detection of IgG and IgM antibodies to 2019 Novel Coronavirus in human whole blood, serum or plasma. Capillary blood was obtained from each patient by fingerstick, captured in a capillary tube and then dispensed to the specimen well of the test cassette. The test uses anti-human IgM antibody (test line IgM), anti-human IgG (test line IgG) and rabbit IgG (control line C) immobilized on a nitrocellulose strip. The test cassette contains colloidal gold conjugated to recombinant COVID-19 antigens (COVID-19 conjugates). When a specimen followed by assay buffer is added to the sample well, IgM and/or IgG antibodies, if present, bind to COVID-19 conjugates making antigen antibodies complex. This complex migrates through nitrocellulose membrane by capillary action. When the complex meets the line of the corresponding immobilized antibody (anti-human IgM and/or anti-human IgG), it is trapped forming a burgundy-colored band which confirms a reactive test result. Absence of a colored band in the test region indicates a non-reactive test result. The use of rapid serological tests on peripheral blood for COVID-19 IgM and IgG detection to all cancer patients was not conceived as a clinical trial but as an operative procedure which was approved by the Hospital according to the current legislation. Informed consent was obtained from the patients before the enrollment. To assess the actual seroprevalence rate and evaluate the clinical performance in the oncological setting of the COVID-19 IgG/IgM Rapid Test Cassette GCCOV-402, a group of patients (all those with positive rapid test and 63 with negative rapid test, randomly selected among those with no previous COVID-19 diagnosis) also underwent a reference validated serological test for SARS-CoV-2 IgM and IgG detection (chemiluminescent immunoassay SARS-CoV-2, Shenzhen YHLO Biotech. Co., Ltd., Shenzhen, China).

### 2.2. Statistical Analysis

Clinicopathological characteristics, together with treatment information were presented using count and percentage for categorical variables, mean or median for continuous variables. Differences between groups were analyzed using Fisher’s exact test or chi-square test for categorical variables and unpaired Student t test, or the Mann–Whitney U test for continuous variables. The 95% confidence intervals (CIs) of the seroprevalence were calculated from binomial probabilities using Miettinen’s exact method. To evaluate consistency of the COVID-19 IgG/IgM Rapid Test Cassette with reference serological test, 2 × 2 tabulation was adopted, and Cohen’s Kappa value was computed. All statistical tests were performed 2-sided at a significance level of α = 0.05. R software (V.3.6.0) and RStudio software (Version 1.2.1335) were used for statistical analyses.

## 3. Results

### 3.1. Study Population

Overall, 949 patients with active cancer undergoing treatment at UNIVPM (532 female and 417 male, median age 67 years (range 18–94)) received a total of 3767 rapid tests (mean number: 3.92, range 1–13). To maintain the Institute as a COVID-free hospital in the study period, the day before the planned hospital access, all patients were advised by phone not to come to the hospital if they had COVID-suspected symptoms or contacts. In addition, on the day planned for anticancer treatment, every patient had to undergo a standard clinical triage procedure which included patient personal and family history for COVID-19 infection and symptoms, vital signs and temperature check. Based on the triage procedure, only a minority of patients reported a previous COVID-19 diagnosis (8; 1%); 117 (12%) patients had performed in the previous months an NPS swab and 47 (5%) referred suspicious symptoms in the last two weeks before treatment; 8 had observed a quarantine period because of infection or suspected contacts. Baseline patient and disease characteristics as well as data on previous COVID-19 history are shown in Table 1. 

### 3.2. Antibodies Seroprevalence and Clinical Performance of the Rapid Serological Test in the Oncological Setting

After performing the GCCOV-402a rapid serological test, 936 patients (98.6%) were IgG-negative and IgM-negative, and only 13 (1.4%) were IgM-positive and/or IgG-positive. In particular, 10 (77% of the total positive) were IgM-negative/IgG-positive, 1 (8%) was IgM-positive/IgG-negative and 2 (15%) were IgM-positive/IgG-positive. Detailed description of IgM-positive and/or IgG-positive patients is reported in Table 2. 

Seven seropositive patients were women, with a median age of 67 (range 45–80). The majority (85%) had metastatic tumors, with no clear prevalence of one tumor type. Nine were receiving systemic chemotherapy, 3 targeted therapy, 2 immunotherapy and 2 endocrine therapy. Of the 13 patients, six (46%) reported a previous COVID-19 diagnosis. 

To determine the actual seroprevalence rate and assess the consistency of the COVID-19 IgG/IgM Rapid Test Cassette with reference serological test, all positive and 63 negative patients (control cohort) were cross-checked. All cross-checked negative patients were randomly selected in the whole cohort among those not reporting a previous COVID-19 diagnosis. Of the negative patients, 41 (65%) were women, and median age was 64 (range 32–89). Thirty-eight (61%) had metastatic tumors. Tumor type distribution was the following: 27 (43%) breast cancers, 20 (32%) gastrointestinal cancers, 6 (10%) lung cancers, 11 (17%) other types. Chemotherapy alone or in combination was the most frequent anticancer treatment (36 patients, 58%). Detailed description of cross-checked seronegative patients is reported in Table 2. 

Interestingly, only 7 out of 13 (54%) were positive at the reference serological test (true positives), while 6 (46%) resulted negative (false positives) suggesting an initial assay’s unspecific reaction. Therefore, antibodies seroprevalence after cross-checking resulted 0.7%. Conversely, all 63 patients who resulted seronegative at the COVID-19 IgG/IgM Rapid Test Cassette were also negative at the reference serological test (100% true negatives, 0% false negatives). Compared with the reference reagent, the COVID-19 IgG/IgM Rapid Test Cassette had a positive predictive value of 54% (95% CI 50–57%) and a negative predictive value of 100% (95% CI 98–100%). The positive agreement was 100% (95%CI 88–100%), the negative agreement was 90% (95% CI 89–90%), and total agreement was 92% (95% CI 91–92%). The kappa value of the consistency analysis was 0.71.

### 3.3. Duration of Immunity and Seroconversion Rate among Cancer Patients

Apart from being postponed at time of first positivity detection, no positive patients on the rapid test had their treatment plan changed. Eight were found positive since first rapid testing, 4 at the second testing and 1 at third testing. The average number of tests for positive patients was 4.31 (range 1–9). Among the 5 positive patients (confirmed after cross-checking) who performed > 1 rapid test and resulted positive for IgG only, 1 performed 2 tests and resulted positive only at the second while the others retained positivity for a median of 46.5 days at time of study cut-off. Among the other patients IgG only-positive to the rapid test, 3 performed >1 rapid test but were not confirmed after cross-checking, while 2 were confirmed but performed only 1 rapid test. The 2 patients who were found positive both for IgM and IgG at the rapid test were not further confirmed after cross-checking and were excluded from the analysis immunity duration; similarly, the patient who was IgM only-positive performed just 1 rapid test and was not confirmed after standard serological test. Results over time of patients positive to the rapid serological test are shown in Table 3.

Looking at seroconversion rate, six out of 8 patients with a previous COVID-19 diagnosis (confirmed by NPS/RT-PCR) resulted seropositive at the GCCOV-402a rapid serological test (rapid test seroconversion rate: 75%). However, only 4 were confirmed IgM/IgG positive after cross-checking (confirmed seroconversion rate: 50%).

## 4. Discussion

COVID-19 disease outbreak between late 2019 and 2020 has dramatically changed people’s everyday life. All countries around the world are engaged in a massive struggle in order to slow down the pandemic and protect the population, in particular the frailest groups [14]. Patients and healthcare workers have been negatively impacted by the ongoing pandemic, not only physically but mentally too [15]. To this regard, the impact of COVID-19 on cancer patients is not clearly defined. Many studies demonstrated a higher risk of infection and developing complications in people with malignancies [1,16,17], while others [3,5] showed a similar incidence of SARS-CoV-2 between people with tumors and healthcare operators. Focusing on Italian real-life experiences, a large study from Aschele et al. [18] who analyzed 59,989 patients from 118 Medical Oncology Units all over Italy between 15 January and 4 May 2020, recorded a 0.68% infection rate among active treatment patients with 77% of them being hospitalized. Bertuzzi et al. [19] reported a 1.8% COVID-19 incidence rate (17/1267); SARS-CoV-2 infected patients required hospitalization in 82% of cases (14/17) with a 29% fatality rate. 

Looking at SARS-CoV-2 antibodies seroprevalence, Zambelli et al. [12] studied cancer patients referred to the Bergamo Hospital, one of the epicenters of COVID-19 pandemic in Italy, and found 31% of them as IgG or IgM positive to COVID-19 using standard serological tests. Similarly, Cabezón-Gutiérreza et al. [20] recorded a 31% COVID-19 IgM/IgG seroprevalence in the city of Madrid, Spain. Contrary, SARS-CoV-2 antibodies were detected only in 2 (3.3%) healthcare operators and 3 patients by Feuderer et al. [13] in Austria, where contagion was not so widespread, between 21 March and 4 June. 

In our study, 13 patients out of 949 (1.4%) have been found positive to GCCOV-402 rapid serological test in the study period. However, after cross-checking with iFlash 3000 chemiluminescence assay for IgM/IgG SARS-CoV-2 antibodies, the GCCOV-402 rapid serological test showed controversial results. In fact, only 7 out of 13 were confirmed as positive at the reference serological test (true positives), thus seroprevalence after cross-checking resulted 0.7%. Positive agreement with standard serological test was 100% (95% CI 88–100%), and negative agreement was 90% (95% CI 89–90%) with a 92% total agreement. Cohen’s kappa was 0.71 [21]. Rapid test negative predictive value was 100%, while GCCOV-402 positive predictive value was only 54% (95% CI 50–57%) compared with standard serological test, leading to an initial overestimation of seroprevalence by the rapid test. Considering the low GCCOV-402a positive predictive value, standard serological test should remain the gold standard to reliably assess patients SARS-CoV-2 immunity. 

The observed seroprevalence rate is quite low compared with data shown by Zambelli et al. in Lombardy [12] but also by Feuderer et al. in Austria [13]. Furthermore, cancer patient seroprevalence in our center was also lower than 2.7% recorded by ISTAT and Italian Ministry of Health in the Marche region [11] though it is uncertain whether it is due to low prevalence of COVID-19 disease in Ancona compared with the rest of our region or because of the measure we took to keep patients safe from contagion. 

The low seroprevalence rate observed in our analysis could be also explained by the different post-infection seroconversion rates in people with malignancies. Liu et al. [22] found that only 72.5% (95% CI 58.0–87.0%) of 40 cancer patients affected by COVID-19 developed SARS-CoV-2 specific antibodies after 21 days from symptoms onset compared to 90.3% (95% CI 88.7–91.8%) in others (*p* < 0.001). IgM prevalence was 20.0% (95% CI 7.0–33.0%) in cancer patients with COVID-19 vs. 31.7% (95% CI 29.3–34.1%) in other patients. Both these values are much lower than previously reported in general population in other single-center studies [20]. On the other hand, Marra et al. [23], analyzing data from 166 patients and healthcare workers in Milan, recently found no differences in seroconversion rates between cancer patients (87.9%) and others (80.5%) (*p* = 0.39), making SARS-CoV-2 immunogenicity still an open issue. Focusing on clinical-pathological features, Cabezón-Gutiérreza et al. [20] demonstrated pneumonia to be the only variable influencing immunity while other factors such as cancer type, stage or treatment did not significantly impact. In addition, patients treated with chemotherapy lacked the ability to produce antibodies more than patients treated with different drugs (immunotherapy, hormone therapy, target therapy), maybe due to chemotherapy immunosuppressive effect. Interestingly, cancer patients treated with drugs other than chemotherapy developed antibodies more than those not receiving any treatment at all, demonstrating that therapeutic approaches different from chemotherapy might also improve immune response to the virus. 

In our case, seroconversion rate of cancer patients resulted 75% using the rapid test and 50% after cross-checking but a comparison cohort is lacking. Patients who performed >1 rapid test retained IgG antibodies positivity for a median of 46.5 days. However, as GCCOV-402a is a qualitative test, it was not possible to detect changes in antibodies levels over time. To note, in our study patients previously affected by COVID-19 were most probably infected between March and June 2020, and their IgG/IgM state before first access as well as their antibody assessment after the end of the study were not available. Therefore, it is not possible to conclude about real duration of immunity in our patients, and studies with a longer follow up are needed. Nevertheless, considering GCCOV-402a easy of execution, relatively low cost and excellent negative predictive value, rapid test may represent a useful tool to follow immune patients over time after IgG/IgM antibodies first detection with standard serological test. 

Anyway, as vaccine distribution has just started, and WHO placed cancer patients in 2nd phase after healthcare workers and the elderly [24,25], although the role of patient immunocompetence is uncertain, having assessed such a low IgG/IgM seroprevalence in our center might help defining where the contagion risk is still high and faster intervention are needed. In addition, as we still lack data about patients’ immunity duration, GCCOV-402a may provide an easy way to determine when a vaccination booster should be considered.

## 5. Conclusions

In conclusion, our results depict the epidemiological reality of COVID-19 among cancer patients in our center. GCCOV-402a high false positive rate and lack of a quantitative assessment of IgG/IgM levels do not render it a reliable tool to determine patients’ level of immunity, and standard serological test should remain the gold standard. On the other hand, given its ease-of-use, GCCOV-402a rapid test may be a useful tool to monitor patients’ immunity over time after SARS-CoV-2 specific antibodies production has been first demonstrated with standard serological test, thus providing data to support and guide vaccination programs. While waiting for large-scale vaccine distribution, oncology departments should continue to ensure the continuity of cancer care by fighting COVID-19 spread. 

## Figures and Tables

**Table 1 jcm-10-01503-t001:** Characteristics of tested patients.

Characteristics	Number (%)
**SARS-CoV-2 rapid tests performed**	3767
**Patients tested**	949
**Average number of tests performed per patient**	3.92
**Median age (range)**	67 (18–94)
**Gender**	
Male	417 (44)
Female	532 (56)
**Previous diagnosis of COVID-19**	
Yes	8 (1)
No	941 (99)
**Previous SARS-CoV-2 swab**	
Performed	117 (12)
Not performed	832 (88)
**Suspicious symptoms in the previous two weeks**	
Yes	47 (5)
No	902 (95)
**Previous quarantine**	
Yes	8 (1)
No	941 (99)
**Body temperature greater than 98.6 °F (>37 °C) at the moment of hospital admission**	
Yes	9 (1)
No	940 (99)
**Rapid test result for antibodies**	
Positive	13 (1)
Negative	936 (99)

**Table 2 jcm-10-01503-t002:** Characteristics of cross-checked seropositive and seronegative patients.

Characteristic	Seropositive Patients (%)	Seronegative Patients (%)	*p* Value ^a^
**Patients**	13 (100)	63 (100)	
IgM positivity	1 (8)	NA	
IgG positivity	10 (77)	NA	
IgM/IgG positivity	2 (15)	NA	
**Median age (range)**	67 (45–80)	64 (32–89)	0.40
**Gender**			
Male	6 (46)	22 (35)	0.44
Female	7 (54)	41 (65)	
**Tobacco smoking**			
Yes	5 (39)	30 (48)	0.70
No	6 (46)	28 (44)	
Unknown	2 (15)	5 (8)	
**Cancer type**			
Breast	2 (15)	27 (43)	0.23
Lung	3 (23)	6 (10)	
GI	5 (38)	20 (32)	
Genitourinary tract	1 (8)	1 (2)	
Others	2 (15)	10 (16)	
**Stage**			
Advanced	11 (85)	38 (60)	0.09
Local/Locally advanced	2 (15)	26 (41)	
**Treatment type ^b^**			
Chemoterapy	9 (69)	36 (57)	0.07
Target terapy	3 (23)	28 (44)	
Immunotherapy	2 (15)	1 (2)	
Endocrine therapy	2 (15)	14 (22)	
**Previous diagnosis of COVID-19**			
Yes	6 (46)	0 (0)	<0.01 ^c^
No	7 (54)	63 (100)	
**Concordance between rapid serological test/reference test**			
Yes (true-positive/true-negative)	7 (54)	63 (100)	<0.01 ^c^
No (false-positive/false-negative)	6 (46)	0 (0)	
**Result of swab performed after seropositivity**			
Positive	0 (0)	NA	NA
Negative	13 (100)	NA	

^a^ Fisher’s exact test or chi-square test comparing proportions between cross-checked seropositive and seronegative patients. P values were calculated excluding unknown values. ^b^ Percentages exceed 100% because some patients were simultaneously receiving more treatments. ^c^ Statistically significant (*p* < 0.05).

**Table 3 jcm-10-01503-t003:** Results over time of patients positive to the rapid serological test.

Patient	Total Rapid Tests	First Seropositivity Date	IgG/IgM Positivity	Last Seropositivity Date	Duration of Seropositivity ^a^	Standard Test Concordance	Previous COVID-19 Diagnosis
1	4 ^a^	08/09/2020	IgG	22/10/2020	44 days	No	No
2	5 ^a^	31/07/2020	IgG	13/09/2020	44 days	Yes	Yes
3	1	22/07/2020	IgM	22/07/2020	\	No	Yes
4	4 ^c^	03/09/2020	IgG	03/09/2020	\	No	No
5	6 ^b^	18/08/2020	IgM/IgG	08/09/2020	21 days	No	No
6	1	27/07/2020	IgG	27/07/2020	\	Yes	No
7	7 ^b^	27/07/2020	IgM/IgG	22/10/2020	87 days	No	No
8	4 ^a^	14/07/2020	IgG	25/08/2020	42 days	Yes	Yes
9	5 ^b^	05/08/2020	IgG	21/10/2020	77 days	No	Yes
10	1	28/07/2020	IgG	28/07/2020	\	Yes	No
11	7 ^a^	10/07/2020	IgG	28/08/2020	49 days	Yes	Yes
12	2 ^b^	21/10/2020	IgG	21/10/2020	\	Yes	Yes
13	10 ^a^	09/07/2020	IgG	13/10/2020	96 days	Yes	No

^a^ Positive since the first test. ^b^ Positive since the second test. ^c^ Positive since the third test. ^d^ Median seropositivity: 46.5 days (only patients confirmed after crosschecking were taken into account for this calculation).

## Data Availability

The datasets used during the present study are available from the corresponding author upon reasonable request.

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
