# Peer review of "Seroprevalence of SARS-CoV-2–Specific Antibodies in Cancer Patients Undergoing Active Systemic Treatment: A Single-Center Experience from the Marche Region, Italy"

_jcm, 2021, doi:10.3390/jcm10071503_

Round 1

Reviewer 1 Report

This article has improved because of the focus on assessment of seroprevalence in their cancer patients in order to add data about COVID19 epidemiology in their region. It will be better if you modify the following.

  1. Please show the result part 3.2 in the Table. Table should present the time course of the positive or negative IgG/IgM of 13 seropositive patients.
  2. Some of the text in the conclusions section need to be moved to discussion section.

Author Response

This article has improved because of the focus on assessment of seroprevalence in their cancer patients in order to add data about COVID19 epidemiology in their region. It will be better if you modify the following.

1-Please show the result part 3.2 in the Table. Table should present the time course of the positive or negative IgG/IgM of 13 seropositive patients.

Thanks for the suggestion. We added a table to clearly describe our results showing changes in our patients immunity over time

2-Some of the text in the conclusions section need to be moved to discussion section.

Thanks for your suggestion, we moved some of the conclusions into the discussion following your advice (lines 346-352) in order to make our conclusions more concise.

Reviewer 2 Report

Thank you for addressing my initial comments. In future please ensure that all line numbers quoted in the comments to the editors line up with the line numbers in the manuscript, as it makes it easier to cross check between the two documents.

Author Response

We really apologize for the inconvenience and confusion we caused. We will surely follow your advice in future.